# The combination of zalfermin and semaglutide has additive therapeutic effects in a diet-induced obese and biopsy-confirmed mouse model of MASH

Jenny Norlin[1☯], Maria Dermit[2☯], Nikos Sidiropoulos[1], Elisabeth D. Galsgaard[1], Henrik H. Hansen[3], Michael Feigh[3], Sanne S. Veidal[1], Markus Latta[1], Emma Henriksson[1], Birgitte Andersen[1]*

1 Global Drug Discovery, Novo Nordisk, Maaloev, Denmark, 2 AI & Digital Research, Research & Early Development, Novo Nordisk Research Centre, Oxford, United Kingdom, 3 Gubra, Hørsholm, Denmark

☯ These authors contributed equally to this work.
* btta@novonordisk.com

## Abstract

Fibroblast growth factor 21 (FGF21) analogs have significant therapeutic potential in metabolic dysfunction–associated steatohepatitis (MASH) but limited body weight effects in patients with MASH. This study investigated the effect of combined treatment with the FGF21 analog zalfermin and the glucagon-like peptide-1 receptor agonist semaglutide on body weight and plasma and liver biochemistry and histology in a mouse model of MASH. Amylin liver nonalcoholic steatohepatitis diet-induced obese-MASH mice with biopsy-confirmed MASH and fibrosis were administered (subcutaneous [SC], daily [QD]) vehicle, zalfermin (0.05 or 0.2 mg/kg), semaglutide (3 or 120 μg/kg), or zalfermin 0.05 mg/kg + semaglutide 3 μg/kg for 8 weeks (n = 11–12 per group). Vehicle-dosed (SC, QD) chow-fed mice served as normal controls (n = 10). Pre- to post-liver biopsy histology was compared for within-subject evaluation of changes in non-alcoholic fatty liver disease Activity Score (NAS), fibrosis stage, and quantitative histology. Additional endpoints included plasma/liver biochemistry and liver RNA sequencing. Combined low-dose zalfermin and semaglutide treatment resulted in super-additive body weight loss (−18%) vs. individual low-dose monotherapies (zalfermin, −6%; semaglutide, −4%) and was equally effective as high-dose zalfermin monotherapy (−16%) and semaglutide (−15%). Low-dose combination therapy promoted greater benefits on transaminases, total cholesterol and triglycerides, NAS, steatosis, and inflammation vs. individual low-dose monotherapies and high-dose semaglutide, and high-dose zalfermin was as effective as the low-dose combination therapy on most endpoints. Combination treatment reduced gene expression markers of fibrosis to a greater degree than monotherapies. In conclusion, combined low-dose zalfermin and semaglutide, as well as high-dose zalfermin, resulted in beneficial

**Data availability statement:** The RNA sequencing data are available in the Gene Expression Omnibus (GEO, https://www.ncbi.nlm.nih.gov/geo/; accession number GSE256063). The code used to generate the results is available in the GitHub repository (https://github.com/novonordisk-research/Zalfermin_liver_RNAseq).

**Funding:** Funded by Novo Nordisk A/S, in accordance with Good Publication Practice (GPP) guidelines (www.ismpp.org/gpp-2022).

**Competing interests:** The study was funded by Novo Nordisk A/S. The funder was involved in the study design, data collection and analysis, decision to publish, and preparation of the manuscript. This does not alter our adherence to PLOS ONE policies on sharing data and materials.

effects on body weight and biochemical and histological endpoints, supporting the clinical development of zalfermin as therapy for patients with MASH.

## Introduction

Metabolic dysfunction–associated steatotic liver disease (MASLD), formerly known as non-alcoholic fatty liver disease (NAFLD), is characterized by hepatic steatosis and at least one cardiometabolic risk factor (e.g., type 2 diabetes, obesity, hyperlipidemia) [1]. MASLD can progress to a more severe form of the disease, metabolic dysfunction–associated steatohepatitis (MASH), formerly known as non-alcoholic steatohepatitis (NASH) [2]. MASH is strongly associated with the development of liver fibrosis, which is the major risk factor for development of cirrhosis, primary liver cancer, and end-stage liver disease. Several molecular targets and drugs are currently being explored, and there is a strong rationale for determining drug combinations that can improve treatment outcomes in MASH [3].

Fibroblast growth factor 21 (FGF21) is a metabolic regulator mainly secreted from the liver [4], which signals through the FGF receptors FGFR1, FGFR2, and FGFR3 but only in the presence of its obligate co-receptor β-klotho (KLB) [5]. In mice, FGF21-receptor complexes are expressed in adipose tissue, liver, pancreas, and specific areas of the central nervous system [4]. In preclinical studies, FGF21 lowered plasma insulin, triglycerides, liver fat, and body weight [6], which led to high expectations for a novel, efficacious treatment for obesity, type 2 diabetes, dyslipidemia, and MASH [7]. Although the clinical efficacy on body weight of FGF21 analogs has been limited, significant and clinically meaningful effects on lipids and MASH parameters have been demonstrated [8–10]. Recently, the long-acting Fc-FGF21 analog efruxifermin established therapeutic potential in MASH [8]. Other long-acting FGF21 analogs, including pegozafermin, BOS-580 (LLL580), and zalfermin, are being investigated in clinical trials for MASH [8,9,11–13]. Zalfermin is a novel, long-acting, fatty acid–modified FGF21 analog for which clinical data support once-weekly dosing [13].

Long-acting glucagon-like peptide-1 (GLP-1) receptor agonists (RAs) are widely used for the management of type 2 diabetes [14], and semaglutide has also been approved for the treatment of obesity [15]. Importantly, semaglutide treatment for 72 weeks significantly improved MASH resolution in patients with MASH with fibrosis stage F2–F3 compared with placebo [16], and semaglutide is currently in phase 3 clinical development for MASH (ESSENCE; NCT04822181) [16]. The complementary modes of action of FGF21 and GLP-1 have led to the notion that combination therapy with FGF21 analogs and GLP-1RAs may have synergistic effects in MASH [17,18]. Accordingly, a phase 2b study (NCT05016882) is in progress to investigate the effect of zalfermin in combination with semaglutide in patients with MASH with fibrosis stage F2–F3 or compensated cirrhosis (stage F4c).

The present preclinical study aimed to characterize the benefits and molecular mechanisms of combination therapy with zalfermin and semaglutide in the amylin

liver nonalcoholic steatohepatitis (AMLN) diet-induced obese (DIO)-MASH mouse, a translational model of biopsy-confirmed MASH and fibrosis [19,20], with emphasis on clinically relevant endpoints and liver transcriptome changes.

## Materials and methods

### Ethics

The Danish Animal Experiments Inspectorate approved all experiments (license #2013-15-2934-00784). All animal experiments conducted were approved by the internal Gubra Animal Welfare Body, were in full compliance with internationally accepted principles for the care and use of laboratory animals, and conformed to the Animal Research: Reporting of In Vivo Experiments (ARRIVE) guidelines.

### Animals

Male C57BL/6J mice (5–6 weeks old) were from Janvier Labs (Le Genest Saint Isle, France) and housed in a controlled environment (12 h light/dark cycle, lights on at 3 a.m., 21±2°C, humidity 50±10%). Each animal was identified by an implantable subcutaneous microchip (PetID Microchip, E-vet, Haderslev, Denmark). Mice had ad libitum access to tap water and chow (3.22 kcal/g, Altromin 1324; Brogaarden, Hoersholm, Denmark) or AMLN diet (4.5 kcal/g; 40 kcal-% fat, of these 22% trans-fat and 26% saturated fatty acids by weight; 22% fructose, 10% sucrose, 2% cholesterol; D09100301, Research Diets, New Brunswick, NJ). Mice were fed chow or AMLN diet for 36 weeks before initiation of treatment (Fig 1A). Animals underwent liver biopsy for assessment of baseline histopathology, as described in detail previously [21]. Animals were single-housed after the operation and allowed to recover for 3 weeks before the start of treatment. Only AMLN DIO-MASH mice with moderate-to-severe steatosis (score ≥2) and fibrosis (stage ≥F1) were included and evaluated using standard clinical biopsy histopathological scoring criteria [2].

### Intervention treatment

AMLN DIO-MASH mice were randomized and stratified to treatment (n = 11–12 per group) based on baseline quantitative alpha-1 type I collagen (Col1a1) staining (primary factor; proportionate [%] area, see below) and body weight (secondary factor). Chow-fed mice served as normal controls (n = 10). AMLN DIO-MASH mice were dosed (subcutaneously [SC], daily [QD]) for 8 weeks with vehicle, zalfermin 0.05 mg/kg (low-dose zalfermin), zalfermin 0.2 mg/kg (high-dose zalfermin), semaglutide 3 µg/kg (low-dose semaglutide), semaglutide 120 µg/kg (high-dose semaglutide), or zalfermin 0.05 mg/kg + semaglutide 3 µg/kg (two separate injections) (low-dose zalfermin-semaglutide). Low-dose zalfermin-semaglutide combination treatment aimed to promote a weight loss equivalent to that attained by individual high-dose monotherapies. For semaglutide, the selected high dose aimed to reflect a clinically relevant dose in patients with MASH but adjusted for shorter circulating half-life in mice compared with humans (7.5 vs. 168 h) [22,23]. Body weight was measured daily over the entire dosing period. Twenty-four-hour food intake was measured daily during the first 14 days of treatment and once weekly thereafter (on treatment days 19, 26, 33, 40, 47, and 53). Animals were terminated by cardiac puncture under isoflurane anesthesia.

### Plasma and liver biochemistry

Four-hour-fasted terminal blood was sampled from the tail vein, kept on ice, and centrifuged (5 min, 4°C; 6000 g) to generate ethylenediaminetetraacetic acid-stabilized plasma. Plasma alanine aminotransferase (ALT), aspartate aminotransferase (AST), triglycerides (TG), total cholesterol (TC), and liver TG and TC levels were determined as described previously [21].

### Liver histology

Baseline and terminal liver biopsy samples (both from the left lateral lobe) were fixed overnight in 4% paraformaldehyde. Liver tissue was paraffin-embedded and sectioned (3 µm thickness). Sections were stained with hematoxylin-eosin (HE),

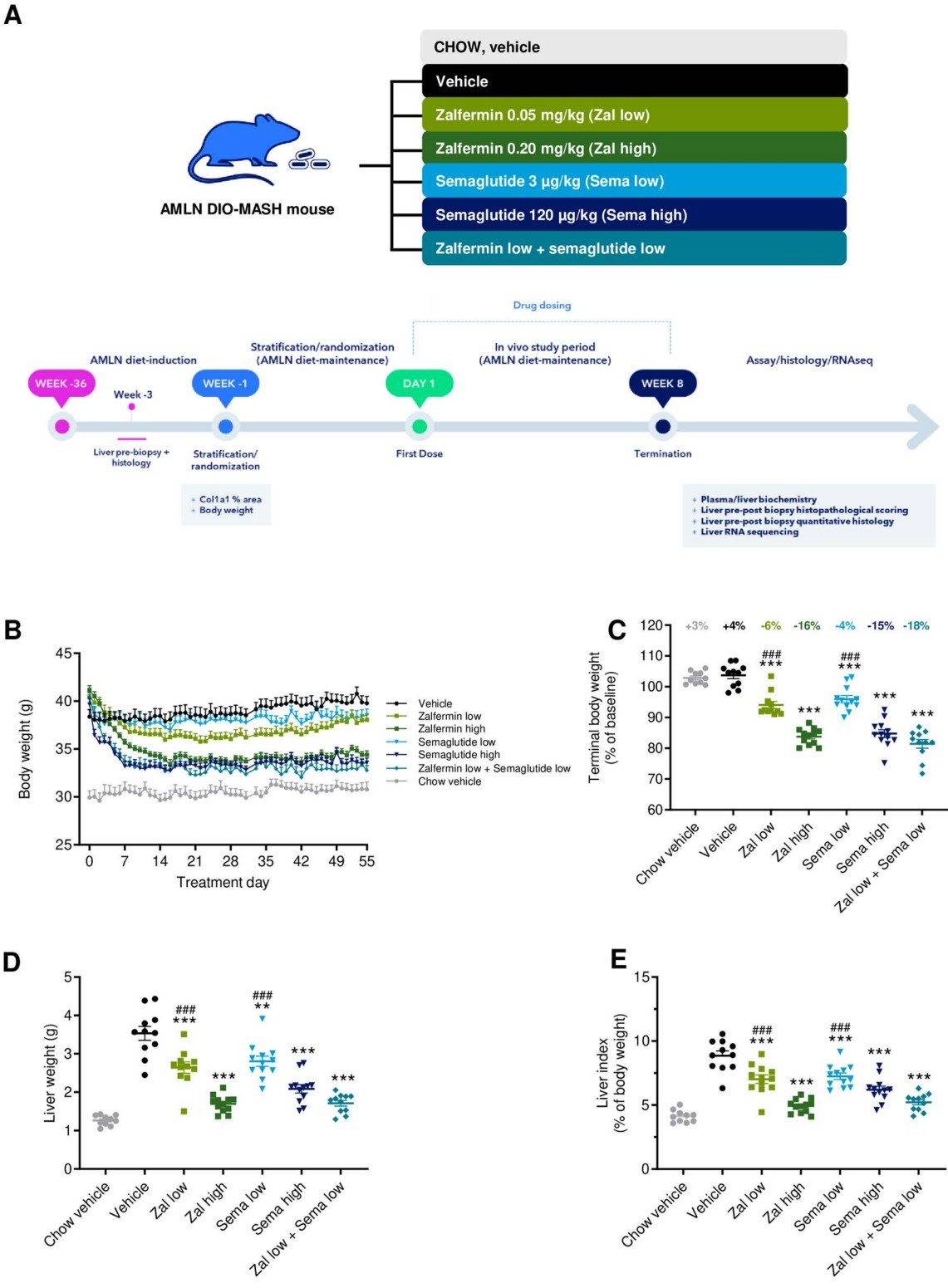

**Fig 1. Low-dose zalfermin-semaglutide combination therapy shows synergistic effects on weight loss and hepatomegaly. (A)** Study outline. AMLN DIO-MASH mice (n = 11–12 per group); vehicle-dosed, chow-fed mice served as normal controls (n = 10). **(B)** Body weight (g) over the entire treatment period. **(C)** Terminal body weight (g). Group-average change in body weight compared with baseline is indicated for each treatment group. **(D)**

Liver weight (g). **(E)** Liver index, calculated as liver weight relative (%) to body weight. Abbreviations: AMLN, amylin liver nonalcoholic steatohepatitis; Col1a1, alpha-1 type I collagen; DIO, diet-induced obese; MASH, metabolic dysfunction–associated steatohepatitis; RNAseq, RNA sequencing; sema, semaglutide; zal, zalfermin. ***$p < 0.001$ vs. DIO-MASH vehicle; **$p < 0.05$ vs. DIO-MASH vehicle; ###$p < 0.001$ vs. zalfermin low + semaglutide low.

picro-Sirius Red (PSR; Sigma-Aldrich, Brøndby, Denmark), anti-cluster of differentiation molecule 11B (anti-CD11b [Integrin αM]) (cat. ab1333357, AbCam, Cambridge, UK), anti-galectin-3 (cat. 125402; Biolegend, San Diego, CA), anti-alpha-smooth muscle actin (α-SMA, cat. ab124964; Abcam, Cambridge, UK), or anti-Col1a1 (cat. 1310−01; Southern Biotech, Birmingham, AL) using standard procedures [21]. Histopathological scoring of steatosis, lobular inflammation, hepatocyte ballooning, and fibrosis stage was performed using the NASH Clinical Research Network scoring system as outlined by Kleiner et al. [2]. Histopathological scores were assessed using an automated deep learning–based image analysis pipeline (Gubra Histopathological Objective Scoring Technique, GHOST), validated previously [24]. Digital image analysis, using the Visiomorph software (Visiopharm, Hørsholm, Denmark), was applied for quantification of markers of inflammation (CD11b, Gal-3), fibrosis (Col1a1), and fibrogenesis (α-SMA) on immunohistochemically stained slides. Area fractions of immunopositive staining were expressed as percentages relative to total parenchymal (fat-free) area by subtracting the corresponding fat area determined on adjacent HE sections.

## RNA sequencing

RNA was extracted from snap-frozen terminal liver samples (15–20 mg fresh tissue, n = 8 per group) and liver transcriptome analysis was performed by RNA sequencing analysis, as previously described in detail [25]. RNA sequence libraries were prepared using NeoPrep (Illumina, San Diego, CA) and the Illumina TruSeq stranded mRNA Library kit for NeoPrep (Illumina) and sequenced using the NextSeq 500 (Illumina) with NSQ 500 hi-Output KT v2 (75 CYS, Illumina). RNA sequencing data were aligned with STAR v2.5.2a using decoy-aware indexes (GRCm38_89). Quality control and alignment statistics were obtained with *MultiQC* (v1.11). All subsequent data processing and analyses were performed in R (v4.2.0) using the *tidyverse* packages (v2.0.0). After loading of RNA sequencing data, *DESeq2* v1.36.0 (default parameters) was used for the quality control and differential expression analyses. Only genes that were expressed in at least three samples with five reads or more were kept for testing of differential expression. Log-fold changes were adjusted using *DESeq2* and Benjamini–Hochberg's method (5% false discovery rate [FDR < 0.05]) was used for multiple testing correction of *p*-values. Overlap of differentially expressed genes was visualized using the *UpSetR* package (v1.4.0). Heatmaps depicting changes in MASH and fibrosis-associated candidate gene expression were prepared using the *pheatmap* package (v1.0.12). Venn diagrams were prepared using the *eulerr* package (v7.0.0). One outlier sample from the DIO-MASH low-dose semaglutide treatment group was excluded after inspection of the clustering results of the variance-stabilized transformed transcriptome data (*DESeq2*'s *vst*) function. After this exclusion, the RNA sequencing dataset had an average 17M mapped reads (standard deviation, 4.5M). Body weight was used as a covariate in the *DESeq2* analysis when performing body weight–independent analyses as previously described [26].

## Statistics

Data are presented as mean ± standard error of the mean. A one-sided Fisher's exact test with Bonferroni correction was used for within-subject comparison of histopathological scores before and after treatment intervention. All other statistical analyses were performed using a one-way analysis of variance (ANOVA) with Dunnett's correction for multiple comparisons. In cases of significantly different standard deviations using a Brown-Forsythe test, Welsh correction ANOVA with Dunnett's T3 correction was applied. The vehicle-dosed chow-fed mouse group was excluded from statistical tests. Superiority to high-dose semaglutide and individual monotherapies was tested using the above statistical methods. A *p*-value less than 0.05 was considered statistically significant.

## Results

The current study aimed to address whether combined treatment with zalfermin and semaglutide has greater benefits over individual monotherapies on metabolic, liver histological hallmarks, and transcriptome signatures in AMLN DIO-MASH mice. With reference to standard practice in clinical trials for MASH, randomization and stratification to treatment was performed based on baseline body weight and liver biopsy histology in AMLN DIO-MASH mice (Fig 1A). Vehicle-subtracted treatment outcomes were calculated to specifically assess for additive/more-than-additive effects of combined low-dose treatment with zalfermin and semaglutide compared with corresponding monotherapies (Table 1).

### Low-dose zalfermin-semaglutide combination therapy has more-than-additive effect on body weight loss

Maximal weight loss (relative to baseline) attained by zalfermin and semaglutide mono- and combination therapy regimens was observed within the first 2 weeks of dosing and was sustained thereafter (Fig 1B). AMLN DIO-MASH mice

**Table 1. Summary of treatment outcomes following treatment with zalfermin and semaglutide, alone and in combination.**

| Endpoint | Vehicle-subtracted treatment outcome | | | | |
|---|---|---|---|---|---|
| | Zalfermin low | Zalfermin high | Semaglutide low | Semaglutide high | Zalfermin low + semaglutide low |
| Body weight (g) | −3.9 ± 0.5### | −8.1 ± 0.4*** | −3.1 ± 0.5### | −7.5 ± 0.6*** | −9.0 ± 0.7ᵃ |
| Body weight (%) | −9.6 ± 1.1*** ### | −19.6 ± 0.8*** | −8.3 ± 1.1*** ### | −18.9 ± 1.3*** | −22.3 ± 1.4*** ᵃ |
| Cumulative food intake (g, days 0–13) | 9.1 ± 1.3*** | 7.5 ± 1.7*** ᵚᵚᵚ | −2.7 ± 0.6# | −9.3 ± 0.8*** | 2.9 ± 2.0ᵚᵚᵚ |
| Liver weight (g) | −0.9 ± 0.1** ### | −1.8 ± 0.1*** | −0.7 ± 0.1* ### | −1.4 ± 0.1*** | −1.8 ± 0.1*** ᵚᵚ ᵃ |
| Liver index (% of body weight) | −1.9 ± 0.3*** ### | −3.9 ± 0.2*** | −1.6 ± 0.3*** ### | −2.7 ± 0.3*** | −3.6 ± 0.2*** ᵚᵚ ᵃ |
| Plasma ALT (U/L) | −99.6 ± 13.1** ### | −178.3 ± 1.7*** | −91.8 ± 12.6** ### | −163.1 ± 7.2*** | −182.8 ± 2.3*** ᵚᵚ |
| Plasma AST (U/L) | −22.6 ± 13.7### | −95.1 ± 3.2*** | −22.5 ± 12.4### | −90.0 ± 6.8*** | −106.3 ± 4.2*** ᵃ |
| Plasma TG (mmol/L) | 0.5 ± 0.0 | 0.4 ± 0.0 | 0.5 ± 0.0 | 0.4 ± 0.1 | 0.4 ± 0.0 |
| Plasma TC (mmol/L) | −1.0 ± 0.5*** | −2.8 ± 0.3* ## | −1.7 ± 0.4** # | −2.2 ± 0.3*** | −3.3 ± 0.2*** ᵚᵚ |
| Liver TG (mg/g liver) | −8.9 ± 2.9### | −37.6 ± 3.6*** | −7.9 ± 5.0### | −27.2 ± 4.5*** | −46.4 ± 4.6*** ᵚᵚ ᵃ |
| Liver TC (mg/g liver) | −0.3 ± 0.4 | −0.4 ± 0.3 | 0.9 ± 0.5 | 0.2 ± 0.6 | 0.1 ± 0.6 |
| NAS | −0.4 ± 0.2### | −1.1 ± 0.0** | −0.3 ± 0.3### | −0.6 ± 0.2 | −1.8 ± 0.3*** ᵚᵚᵚ ᵃ |
| Fibrosis score | 0.2 ± 0.2 | 0.2 ± 0.1 | 0.2 ± 0.2 | 0.2 ± 0.2 | 0.1 ± 0.1 |
| Steatosis score | 0.0 ± 0.0### | −1.0 ± 0.0*** ᵚᵚᵚ | −0.1 ± 0.1### | −0.4 ± 0.2** | −1.0 ± 0.1*** ᵚᵚᵚ ᵃ |
| Lobular inflammation score | −0.3 ± 0.2 | 0.0 ± 0.0 | −0.1 ± 0.2 | −0.1 ± 0.2 | −0.7 ± 0.2* ᵚ ᵃ |
| Hepatocyte ballooning score | −0.1 ± 0.0 | −0.1 ± 0.0 | −0.1 ± 0.1 | −0.1 ± 0.0 | −0.1 ± 0.1 |
| Lipid (% area, HE) | −9.7 ± 1.1*** ### | −21.5 ± 0.9*** ᵚᵚᵚ | −7.5 ± 1.3*** ### | −15.0 ± 1.5*** | −20.2 ± 0.7*** ᵚᵚ ᵃ |
| CD11b (% area, fat-free) | −0.4 ± 0.1 | −0.7 ± 0.1* | −0.4 ± 0.2 | −0.5 ± 0.2 | −0.7 ± 0.2* |
| Gal-3 (% area, fat-free) | −3.3 ± 0.5*** | −4.4 ± 0.2*** | −2.2 ± 0.4** | −3.6 ± 0.6*** | −3.5 ± 0.5*** |
| Col1a1 (% area, fat-free) | −4.3 ± 1.7 | −1.7 ± 0.8 | −2.0 ± 1.4 | −1.6 ± 1.5 | −3.7 ± 0.8 |
| α–SMA (% area, fat-free) | −3.3 ± 0.6*** | −4.8 ± 0.3*** | −2.5 ± 0.5** | −4.0 ± 0.4*** | −4.2 ± 0.4*** |

Data are expressed as change in endpoint (mean ± standard error of the mean) vs. DIO-MASH vehicle controls (vehicle-subtracted treatment outcome). AMLN DIO-MASH mice (n = 11–12 per group) were administered vehicle (SC, QD), zalfermin 0.05 mg/kg (zalfermin low), zalfermin 0.2 mg/kg (zalfermin high), semaglutide 3 µg/kg (semaglutide low), semaglutide 120 µg/kg (semaglutide high), or zalfermin 0.05 mg/kg + semaglutide 3 µg/kg (zalfermin low + semaglutide low) for 8 weeks. The vehicle-dosed chow-fed mouse group (Chow vehicle) was excluded from the analysis.

Abbreviations: α-SMA, alpha-smooth muscle actin; AMLN, amylin liver nonalcoholic steatohepatitis; ALT, alanine aminotransferase; AST, aspartate aminotransferase; CD11b; cluster of differentiation molecule 11B (Integrin αM); Col1a1, alpha-1 type I collagen; Gal-3, galectin-3; HE, hematoxylin-eosin; NAS, Non-Alcoholic Fatty Liver Disease Activity Score; PSR, picro-Sirius Red; SEM, standard error of the mean; TC, total cholesterol; TG, triglycerides.

*$p < 0.05$, **$p < 0.01$, ***$p < 0.001$ vs. DIO-MASH vehicle; #$p < 0.05$, ##$p < 0.01$, ###$p < 0.001$ vs. zalfermin low + semaglutide low; ᵚ$p < 0.05$, ᵚᵚ$p < 0.01$, ᵚᵚᵚ$p < 0.001$ vs. semaglutide high; ᵃadditive or more-than-additive effect compared with sum of zalfermin low and semaglutide low monotherapies.

showed a modest but significant sustained body weight loss after 8 weeks of monotherapy with low-dose zalfermin (−5.9±1.2%, $p<0.001$ vs. DIO-MASH vehicle) and low-dose semaglutide (−4.0±1.1%, $p<0.001$ vs. DIO-MASH vehicle) relative to baseline. Compared with low-dose monotherapies, combined low-dose zalfermin-semaglutide treatment resulted in more-than-additive weight loss (−18.6±1.4%, $p<0.001$ vs. DIO-MASH vehicle, Figs 1B and 1C, Table 1). The magnitude of weight loss attained by low-dose zalfermin-semaglutide combination treatment was comparable to monotherapy with high-dose semaglutide (−15.3±1.3%, $p<0.001$ vs. DIO-MASH vehicle; $p>0.05$ vs. combination treatment) and high-dose zalfermin (−16.3±0.8%, $p<0.001$ vs. DIO-MASH vehicle; $p>0.001$ semaglutide low), respectively. High-dose, but not low-dose, semaglutide monotherapy reduced cumulative food intake vs. DIO-MASH vehicle ($p<0.001$, S1 Fig). In contrast, both low- and high-dose zalfermin monotherapy significantly increased cumulative food intake (both $p<0.001$ vs. DIO-MASH vehicle, S1 Fig). Cumulative food intake was unaffected following low-dose zalfermin-semaglutide combination treatment ($p>0.05$ vs. DIO-MASH vehicle, S1 Fig).

### Low-dose zalfermin-semaglutide combination therapy has added beneficial effects on hepatomegaly, transaminases, and lipid biochemical markers

The low-dose zalfermin-semaglutide combination treatment had additive benefits on hepatomegaly in AMLN DIO-MASH mice. Accordingly, low-dose zalfermin-semaglutide combination treatment resulted in more than a 50% reduction in liver weight compared with vehicle controls (1.7±0.1 g vs. 3.5±0.2 g, $p<0.001$, Fig 1D). Monotherapies showed dose-dependent effects on liver weight (low-dose zalfermin, 2.6±0.2 g, $p<0.01$; high-dose zalfermin, 1.7±0.1 g, $p<0.001$ vs. DIO-MASH vehicle; low-dose semaglutide, 2.8±0.1 g, $p<0.05$; high-dose semaglutide 2.1±0.1 g, $p<0.001$ vs. DIO-MASH vehicle, Fig 1D). Liver index was also additively lowered by low-dose zalfermin-semaglutide combination treatment (5.2±0.2%) compared with DIO-MASH vehicle controls (8.9±0.4%, $p<0.001$, Fig 1E, Table 1). In comparison, improvements in the liver index were less pronounced following low-dose zalfermin (7.0±0.3%, $p<0.001$ vs. DIO-MASH vehicle) and low-dose semaglutide monotherapy (7.3±0.3%, $p<0.001$ vs. DIO-MASH vehicle). Further reductions were achieved with high-dose zalfermin (4.9±0.1%, $p<0.001$ vs. DIO-MASH vehicle) and semaglutide monotherapy (6.2±0.3%, $p<0.001$ vs. DIO-MASH vehicle).

Plasma transaminases were significantly elevated in AMLN DIO-MASH mice (Figs 2A and 2B). Notably, AMLN DIO-MASH mice achieved near-normal plasma ALT levels upon low-dose combination treatment with zalfermin and semaglutide (Fig 2A), reflecting near-additive effects of the treatment regimen (Table 1). Both high-dose monotherapies also promoted substantial reductions in plasma ALT levels. Low-dose zalfermin-semaglutide combination treatment, but not low-dose monotherapies, also significantly reduced plasma AST levels (Fig 2B). The effect of combined low-dose combination treatment was more than additive (Table 1). The efficacy of low-dose zalfermin-semaglutide combination treatment on plasma AST levels was comparable to high-dose zalfermin and semaglutide (Fig 2B).

Plasma TG concentrations were marginally reduced by the high-fat and cholesterol-rich AMLN diet. Plasma TG levels did not change further due to treatments (Fig 2C). In contrast, hypercholesterolemia and elevated liver TG levels in AMLN DIO-MASH mice were markedly lowered by low-dose zalfermin-semaglutide combination treatment ($p<0.001$). The combination regimen demonstrated significantly larger reductions in these two lipid markers compared with high-dose semaglutide monotherapy and was more than additive for liver TG reductions of the two low-dose monotherapies ($p<0.01$ for both endpoints, Figs 2D and 2E, Table 1). High-dose zalfermin was comparable to combination treatment on liver TG, while high-dose semaglutide and zalfermin monotherapy demonstrated comparable therapeutic efficacy on plasma TC. Less-pronounced effects on plasma TC were noted for low-dose monotherapies compared with DIO-MASH controls (zalfermin low-dose, $p<0.05$; semaglutide low-dose, $p<0.01$, Fig 2D). Also, low-dose monotherapies did not improve liver TG levels (Fig 2E). Elevated liver TC levels in AMLN DIO-MASH mice were unaffected by treatments (Fig 2F).

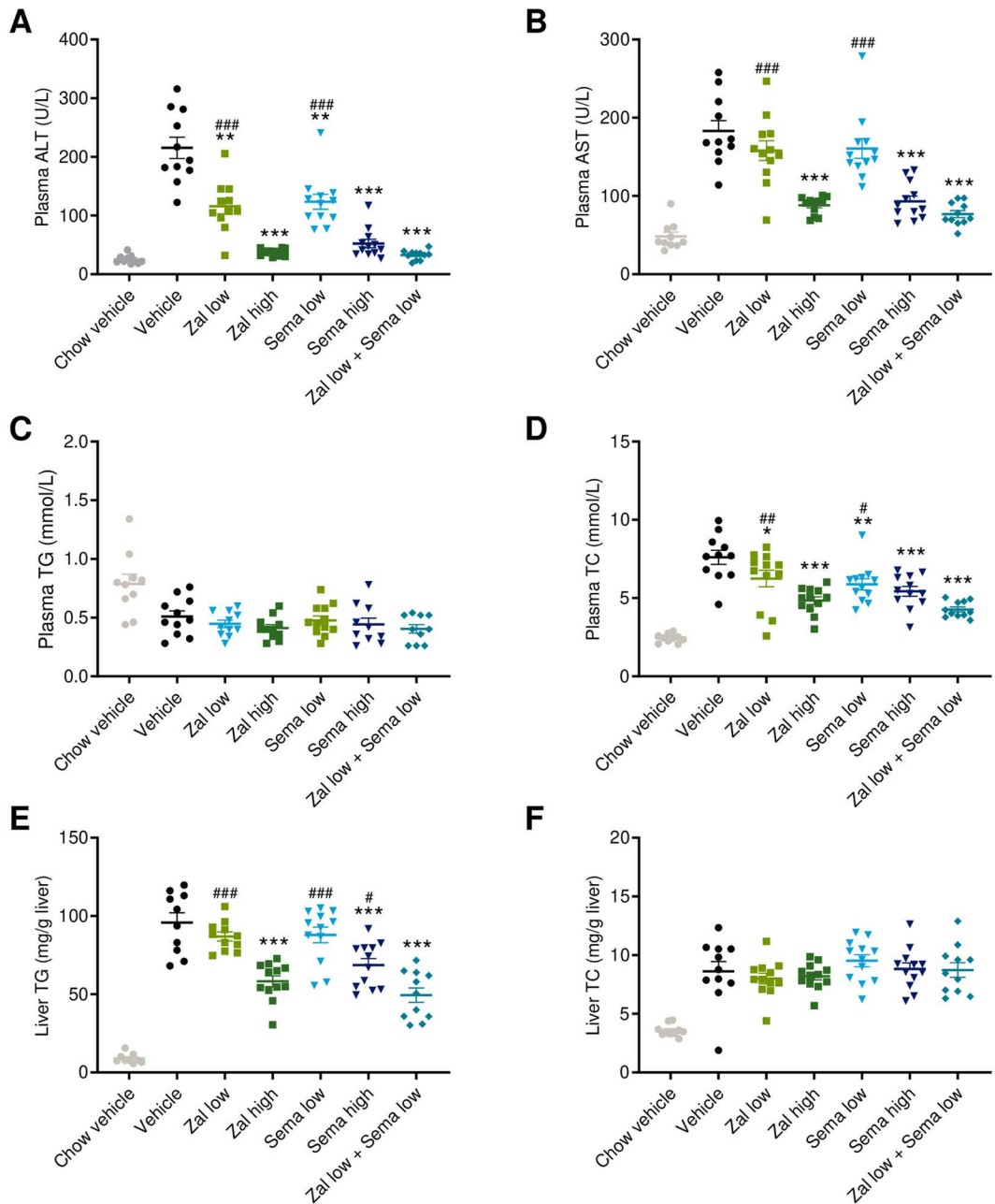

**Fig 2. Low-dose zalfermin-semaglutide combination therapy shows additive effects on plasma and liver biochemical markers. (A)** Plasma ALT (U/L). **(B)** Plasma AST (U/L). **(C)** Plasma TG (mmol/L). **(D)** Plasma TC (mmol/L). **(E)** Liver TG (mg/g liver). **(F)** Liver TC (mg/g liver). Abbreviations: ALT, alanine aminotransferase; AST, aspartate aminotransferase; DIO, diet-induced obesity; MASH, metabolic dysfunction–associated steatohepatitis; sema, semaglutide; TC, total cholesterol; TG, triglycerides; zal, zalfermin. $*p < 0.05$, $**p < 0.01$, $***p < 0.001$ vs. DIO-MASH vehicle; $#p < 0.05$, $##p < 0.01$, $###p < 0.001$ vs. zalfermin low + semaglutide low.

## Low-dose zalfermin-semaglutide combination treatment has added beneficial effects on histological hallmarks of MASH

The effect of zalfermin and semaglutide was evaluated on liver semiquantitative and quantitative liver histological outcomes. Chow controls had normal liver histology. For AMLN DIO-MASH mice, treatment groups were randomized and

stratified to obtain comparable severity of MASH (NAFLD activity score [NAS] 5–6) and fibrosis (stage F2–F3) at baseline (S2 Fig). Treatment outcomes on these clinically derived semiquantitative endpoints were determined by evaluating within-subject pre- to post-liver biopsy histopathological scores (S2 Fig).

Differential histological efficacy profiles were observed for zalfermin and semaglutide monotherapy. Accordingly, of the two, only high-dose zalfermin treatment led to significant improvement in NAS (1 point, 12/12 mice, $p < 0.001$ vs. DIO-MASH vehicle), being largely driven by robust and consistent reductions in steatosis scores (Figs 3A and 3C). Zalfermin and semaglutide alone had no significant effect on lobular inflammation (Fig 3D). Low-dose zalfermin-semaglutide combination treatment resulted in further robust histological benefits, as indicated by improvements in NAS (1 point, 4/11 mice, $p < 0.01$; ≥2 points, 6/11 mice, $p < 0.05$, Fig 3A), which were explained by combined more-than-additive benefits on both steatosis scores (1 point, 9/11 mice, $p < 0.001$; 2 points, 1/11 mice, $p > 0.05$, Fig 3C, Table 1) and lobular inflammation scores (1 point, 4/11 mice, $p < 0.05$; 2 points, 2/11 mice, $p > 0.05$, Fig 3D, Table 1). Hepatocyte ballooning was marginal in AMLN DIO-MASH mice and unaffected by treatments (Fig 3E). Fibrosis stage worsened in all AMLN DIO-MASH mouse groups over the 8-week study period, with an incidence rate ranging from 27% to 42% (Fig 3B). Treatments for 8 weeks did not improve fibrosis scores (Fig 3B).

A series of histomorphometric analyses were performed to confirm histological benefits of low-dose zalfermin-semaglutide combination treatment (Fig 4, S3 Fig). Reduced steatosis severity was corroborated by quantitative digital image analysis of HE-stained sections used for pre- to post-steatosis scoring. Using this approach, we detected significantly reduced fractional (%) area of lipids following monotherapy with low-dose zalfermin or semaglutide (both $p < 0.001$ vs. DIO-MASH vehicle), emphasizing the greater sensitivity of histomorphometry compared with semiquantitative scoring modules. Consistent with further improved steatosis scores, low-dose zalfermin-semaglutide combination treatment reduced liver fat content more than low-dose monotherapies, indicating more-than-additive effects of the low-dose combination regimen (Fig 4A, Table 1). The magnitude of quantitative liver fat depletion afforded by low-dose zalfermin-semaglutide combination treatment was comparable to high-dose monotherapies (Fig 4A). Immunohistochemical analyses were performed to evaluate quantitative effects on standard molecular markers of inflammation (CD11b, Gal-3), fibrosis (Col1a1), and stellate cell activation/fibrogenesis (α-SMA). Overall, low-dose combination treatment reduced parenchymal (fat-free) expression of CD11b, Gal-3, and α-SMA (all $p < 0.001$ vs. DIO-MASH vehicle) to a similar degree to individual monotherapies (Figs 4B, 4C, and 4E). As for fibrosis scores, treatments had no significant effect of quantitative Col1a1 histology (Fig 4D).

Taken together, our data demonstrate that low-dose zalfermin and semaglutide have added therapeutic benefits on body weight, plasma liver enzymes, lipid biochemistry, hepatic steatosis, and liver inflammation in DIO-MASH mice. To further investigate the underlying molecular effects, we evaluated liver transcriptome changes for gene-expression programs associated with these treatment outcomes.

### Low-dose zalfermin-semaglutide combination treatment partially normalizes hepatic transcriptome signatures in DIO-MASH mice

High-quality transcriptome profiles were obtained from liver samples across the experimental groups (Fig 5, S4 Fig). A principal component analysis (PCA) of the 500 most variable genes across all samples indicated clear group separation of differentially expressed genes (DEGs). Most of the variance in the hepatic transcriptome data set was contributed by the disease phenotype (DIO-MASH vehicle vs. chow vehicle, n = 7,758 DEGs, Figs 5A and 5B). All treatments promoted a rightward shift in the PCA plot (i.e., toward chow-fed vehicle controls), indicating gradual normalization of the hepatic global gene expression profile. The rightward shift was most pronounced for high-dose monotherapies and low-dose zalfermin-semaglutide combination therapy (Fig 5A). The total number of DEGs in the treatment groups vs. vehicle-dosed DIO-MASH mice were as follows: low-dose zalfermin (n = 221), low-dose semaglutide (n = 0), low-dose zalfermin-semaglutide combination (n = 3,323), high-dose zalfermin (n = 2,766), and high-dose semaglutide (n = 649) (Fig 5B, S4 Fig). It is noteworthy that the number of DEGs in the low-dose zalfermin-semaglutide group was 15-fold higher compared

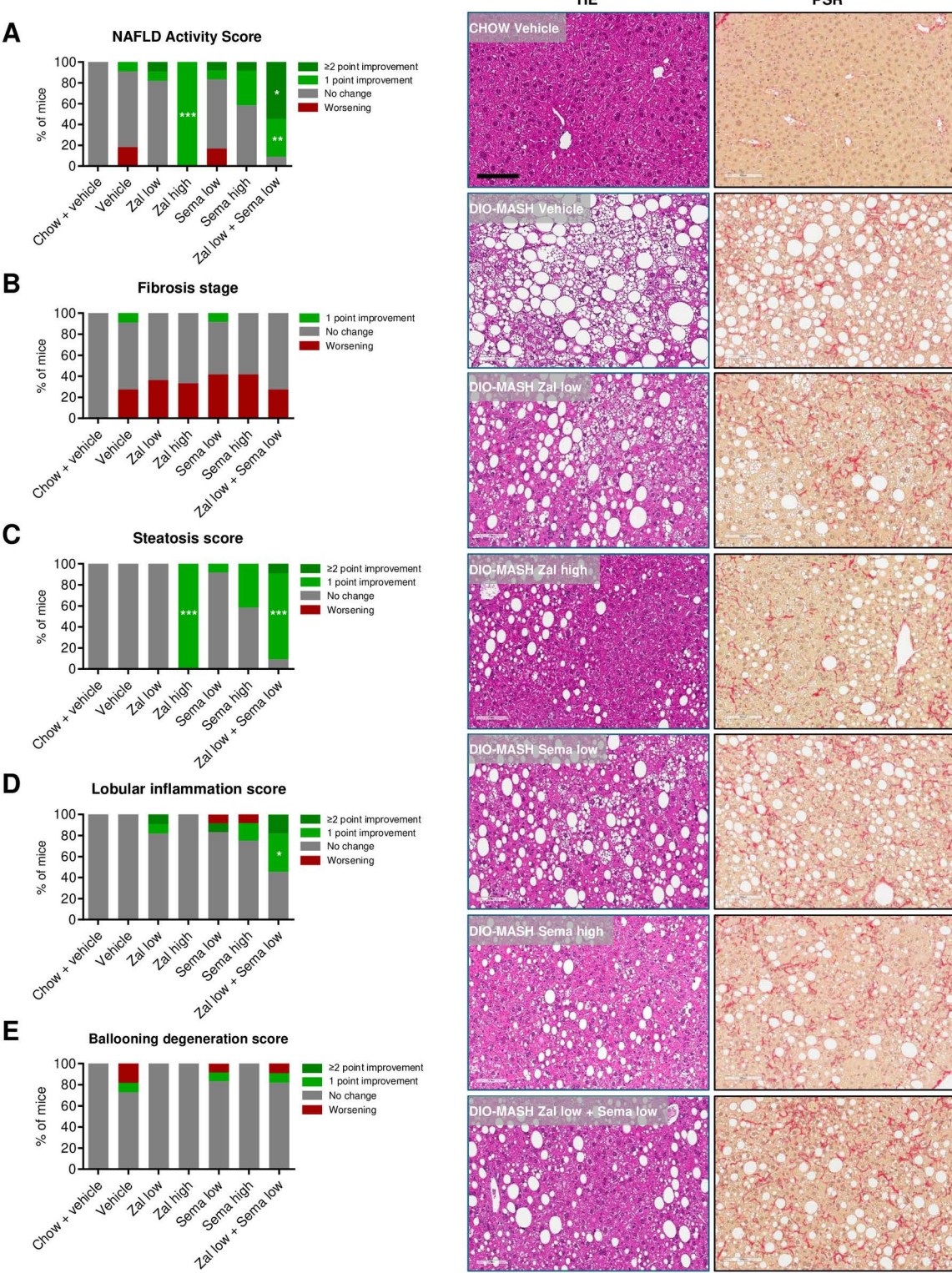

**Fig 3. Low-dose zalfermin-semaglutide combination therapy shows additive effects on liver histopathological hallmarks. (A)** NAS. **(B)** Fibrosis stage. **(C)** Steatosis score. **(D)** Lobular inflammation score. **(E)** Ballooning degeneration score. Abbreviations: DIO, diet-induced obesity; HE, hematoxylin and eosin; MASH, metabolic dysfunction–associated steatohepatitis; NAFLD, Non-Alcoholic Fatty Liver Disease; NAS, Non-Alcoholic Fatty Liver Disease Activity Score; PSR, picro-Sirius Red; sema, semaglutide; zal, zalfermin. $*p < 0.05$, $**p < 0.01$, $***p < 0.001$ vs. DIO-MASH vehicle. Right panels:

Representative photomicrographs showing marked reductions in steatosis and inflammation (HE staining), whereas fibrosis (PSR staining) appears unchanged after low-dose combination therapy with zalfermin and semaglutide. Magnification 20×; scale bar, 100 μm.

with summation of the corresponding low-dose monotherapy groups, indicating more-than-additive effects at the hepatic transcriptome level.

The expression profile of FGF21 and its cognate receptors was also investigated. DIO-MASH mice demonstrated significantly increased *Fgf21* expression and reduced *Ffgr4* and *Klb* expression compared with chow-fed controls (Fig 5C). Hepatic *Fgfr1*, *Fgfr2*, and *Fgfr3* expression was unaltered in DIO-MASH mice. Only low-dose zalfermin-semaglutide combination treatment and high-dose zalfermin monotherapy partially reversed *Ffgr4* expression. Low-dose combination treatment also significantly increased hepatic *Klb* expression relative to DIO-MASH vehicle controls (Fig 5C). A similar effect was observed for high-dose zalfermin and semaglutide, but not for low-dose monotherapies.

**Changes in hepatic transcriptome signatures are partly driven by weight loss–independent effects of low-dose zalfermin-semaglutide combination treatment**

To obtain further resolution of hepatic transcriptome regulations, RNA sequencing data were probed for candidate genes linked to MASH and fibrosis. Consistent with histological hallmarks of MASH and fibrosis, DIO-MASH mice showed widespread regulations in candidate gene markers of liver metabolism, immune function, and extracellular matrix (ECM) organization (Fig 6A). Overall, low-dose zalfermin-semaglutide combination treatment reversed changes in all disease-associated gene categories investigated. Changes in liver metabolic markers suggest that low-dose combination treatment improves intrahepatic handling of carbohydrates (upregulation: *Irs1, Khk, Pygl, Slc2a4*; downregulation: *Mtor*), bile acids (upregulation: *Abcb11, Slc22a7, Slc27a5, Slco1a4*), and lipids (upregulation: *Apoa1, Apoa5, Apoc3, Hmgcr, Ppard, Scarb1, Thrb, Vldlr*; downregulation: *Cd36, Pparg*) (Fig 6A). Only low-dose combination treatment increased gene expression markers associated with endoplasmic reticulum stress (*Atf4, Atf6, Ern1*), displaying a similar profile to chow mice. Notably, low-dose combination treatment suppressed expression of nearly all (22/30) gene markers of ECM remodeling. These included all major collagen subtypes (*Col1a1, Col3a1, Col4a1, Col5a1/2/3, Col6a1/2/3*) as well as several isoforms of matrix metalloproteinases and tissue inhibitors of matrix metalloproteinases, implying reduced pro-fibrogenic activity of the combination treatment (Fig 6A). In comparison, pronounced effects were also observed for gene markers of the immune system (downregulation: *CCl2, Ccr1/2/5, Cysltr1*; upregulation: *Nr1h3, Traf6*). Whereas the above changes in candidate genes were comparable overall to high-dose zalfermin treatment, fewer candidate genes were differentially regulated by high-dose semaglutide.

Considering that both zalfermin and semaglutide promoted weight loss in DIO-MASH mice, and body weight loss is limited in patients with MASH treated with FGF21 analogs, the body weight loss–independent benefits of the treatments on transcriptome signatures in DIO-MASH mice were investigated. To this end, the differential gene expression analyses were adjusted for body weight (Fig 6B, S5 Fig). Compared with chow-fed controls, candidate genes involved in liver metabolism, inflammation, and fibrogenesis remained significantly regulated in DIO-MASH control mice independent of body weight, likely signifying re-routing of hepatic metabolism necessary to manage dysmetabolic and proinflammatory effects of the high-fat/fructose diet. While treatment-induced changes in liver metabolic markers, inflammation, and endoplasmic reticulum stress were largely explained by weight loss, a subset of ECM-related genes remained suppressed after adjusting for body weight in DIO-MASH mice. This only applied to low-dose zalfermin-semaglutide combination treatment (downregulation: *Col3a1, Col4a1, Col5a1/2, Loxl2*) and monotherapy with low-dose zalfermin (downregulation: *Col5a2*) and high-dose zalfermin (downregulation: *Col1a1, Col1a2, Col3a1, Col4a1, Col5a1/2/3, Loxl2, Pdgfa, Timp1*). Hence, low-dose zalfermin-semaglutide combination treatment promotes more

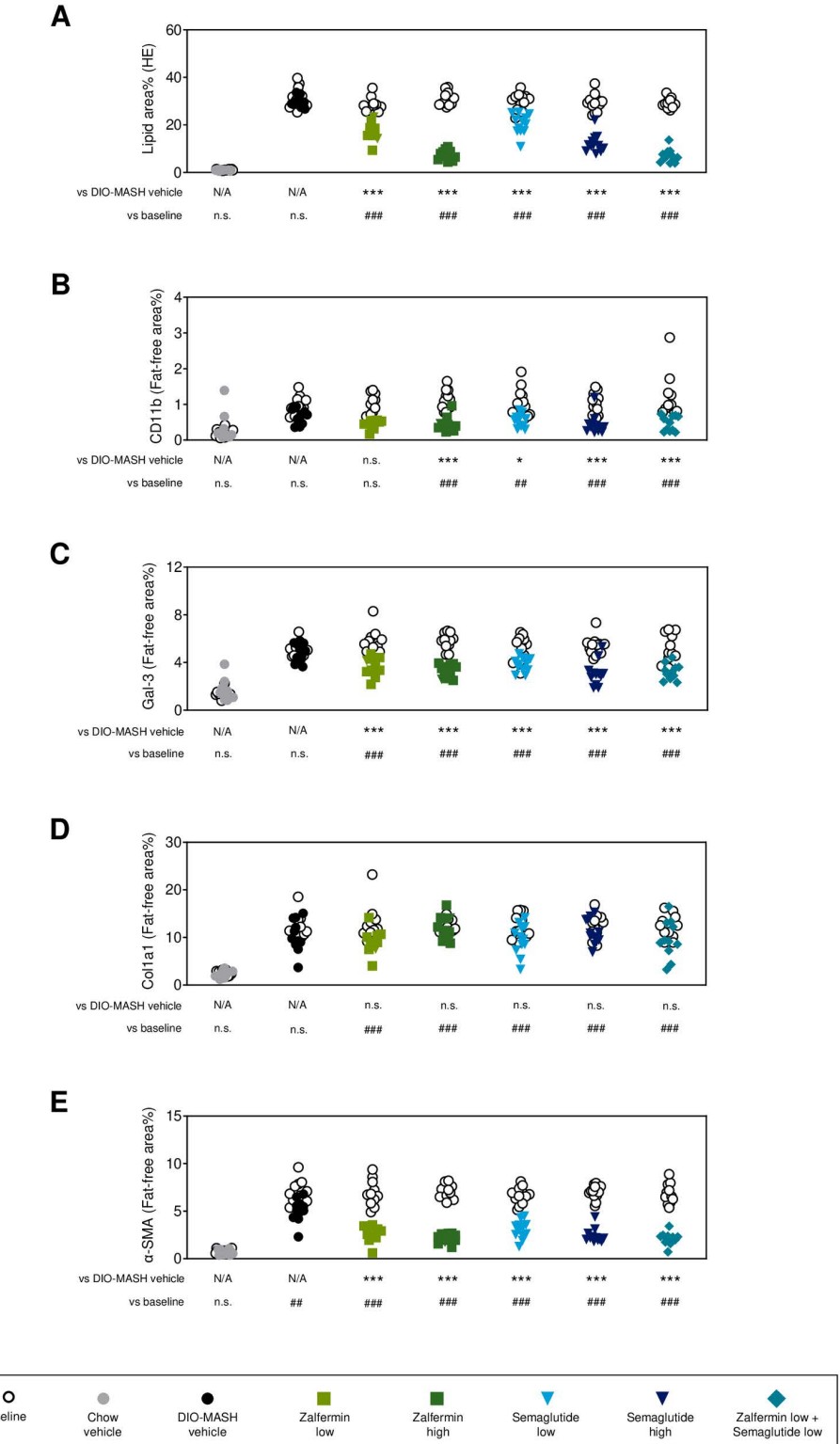

**Fig 4. Low-dose zalfermin-semaglutide combination therapy shows synergistic effects on quantitative histological markers of MASH. (A)** Fat % area (HE staining). **(B)** CD11b. **(C)** Gal-3. **(D)** Col1a1. **(E)** α-SMA. Area fractions of CD11b, Gal-3, Col1a1, and α-SMA immunopositive staining were expressed relative (%) to total parenchymal area by subtracting corresponding fat area determined on adjacent HE sections. Abbreviations: α-SMA,

alpha-smooth muscle actin; CD11b; cluster of differentiation molecule 11B (Integrin αM); Col1a1, alpha-1 type I collagen; DIO, diet-induced obesity; Gal-3, galectin-3; HE, hematoxylin and eosin; MASH, metabolic dysfunction–associated steatohepatitis; N/A, not applicable; n.s., not significant. *$p < 0.05$, ***$p < 0.001$ vs. DIO-MASH vehicle; ##$p < 0.01$, ###$p < 0.001$ vs. baseline.

robust suppression of ECM-associated genes than low-dose zalfermin monotherapy. High-dose zalfermin presented the broadest body weight–independent downregulation of ECM-associated genes. Semaglutide monotherapy did not show any body weight–independent effects on gene expression in this study of 8-week intervention treatment.

## Discussion

The present preclinical study characterized the effects of zalfermin, a novel long-acting FGF21 analog, alone and in combination with semaglutide in the AMLN DIO-MASH mouse, a well-established biopsy-confirmed model of MASH and fibrosis [19,20]. We report additive benefits of combined low-dose zalfermin and semaglutide treatment. Notably, the efficacy of low-dose zalfermin-semaglutide combination therapy on metabolic, biochemical, and histological endpoints was greater than or equal to corresponding high-dose monotherapies. Hepatic transcriptome analyses suggest that effects of zalfermin mono- and combination treatment were partly driven by weight loss–independent molecular mechanisms. Collectively, the current study supports a clinical rationale for development of zalfermin as combination therapy or monotherapy in patients with MASH.

Zalfermin and semaglutide monotherapies showed comparable dose-dependent outcomes on most endpoints in AMLN DIO-MASH mice with biopsy-confirmed MASH. Both compounds consistently reduced body weight and improved biochemical and histological hallmarks of MASH after 8 weeks of treatment, but without reducing fibrosis severity in AMLN DIO-MASH mice. While the benefits of high-dose zalfermin and semaglutide monotherapy on liver histology were predominantly driven by improvements in steatosis, both compounds also reduced standard histological markers of hepatic inflammation and stellate cell activation. The efficacy profile of high-dose semaglutide is in line with previous studies in AMLN DIO-MASH mice [26] and Gubra-Amylin NASH (GAN) DIO-MASH mice [24], a closely related model with similar disease phenotype [27]. Whereas FGF21 analogs have previously been tested in various DIO/DIO-MASLD mouse models [28], only a single efficacy study has so far been reported in a translational model of fibrosing MASH. Accordingly, the FGF21 analog PF-05231023 (10 mg/kg, SC, biweekly dosing) lowered body weight (−7%) in addition to improving liver pathology, including NAS and fibrosis score, in GAN DIO-MASH mice after 12 weeks of treatment [29]. Although differences in study designs should be taken into account, the present study suggests greater efficacy of zalfermin on body weight (−16%) with comparable benefits on MASH endpoints using a substantially lower dose (0.2 mg/kg, SC, QD) and shorter treatment duration (8 weeks). While transcriptome signatures suggested beneficial changes in ECM gene expression patterns following zalfermin combination and monotherapy, the relatively short treatment period could be sufficient to suppress fibrogenesis activity, although longer treatment duration is likely necessary to also clear collagen deposited before treatment intervention. The high-dose zalfermin and the low-dose combination treatment presented with significant reduction in ECM signature even after correcting for body weight loss.

This preclinical study is the first to demonstrate significant benefits of FGF21 analog and GLP-1RA combination therapy in the context of MASH. Outcomes of combined low-dose zalfermin-semaglutide therapy were superior to individual low-dose monotherapies on almost all endpoints evaluated in AMLN DIO-MASH mice. It is noteworthy that additive or more-than-additive effects were observed for improvements in body weight, hepatomegaly, biochemistry (plasma cholesterol, plasma ALT, liver TG), and histology (NAS, steatosis score, lobular inflammation score, lipid % area). The therapeutic benefits of high-dose semaglutide treatment in AMLN and GAN DIO-MASH mice are in close agreement with a recent clinical phase 2 trial in patients with noncirrhotic MASH [16]. Currently, semaglutide is in phase 3 clinical development for MASH (ESSENCE; NCT04822181) and has recently received accelerated FDA approval for MASH with

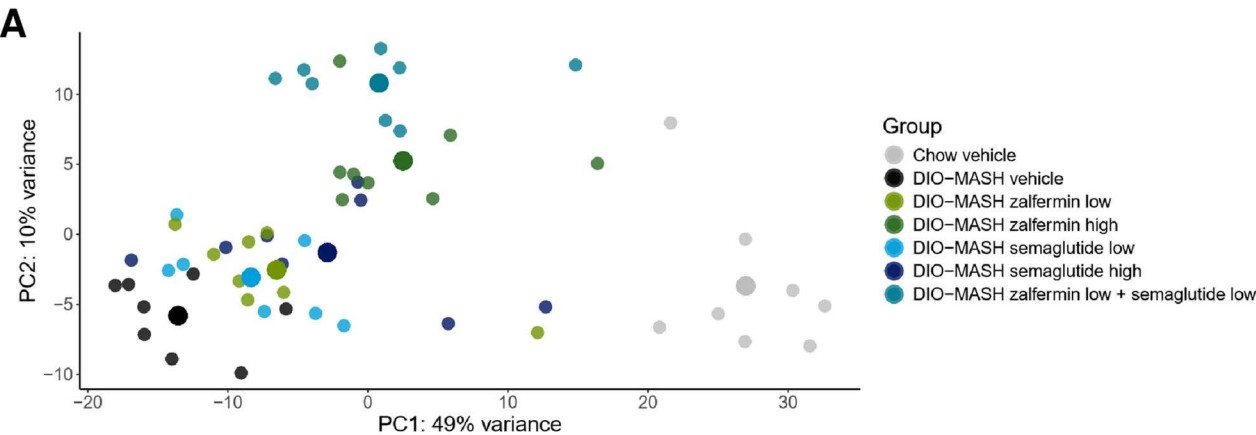

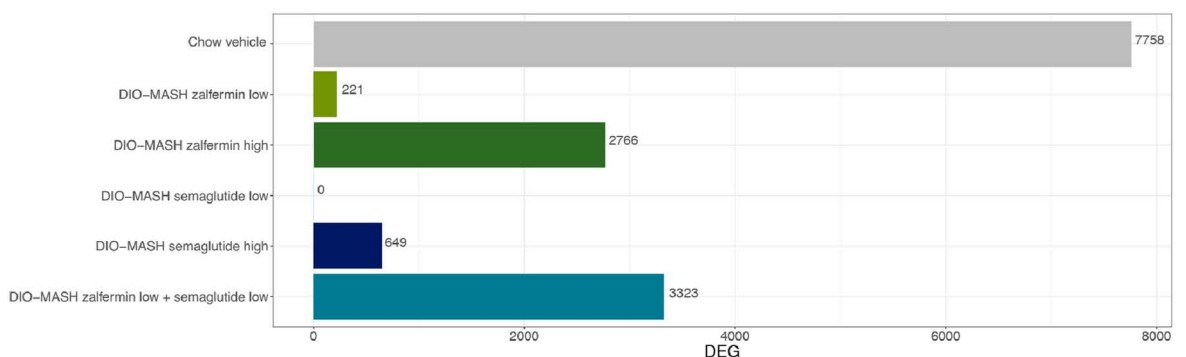

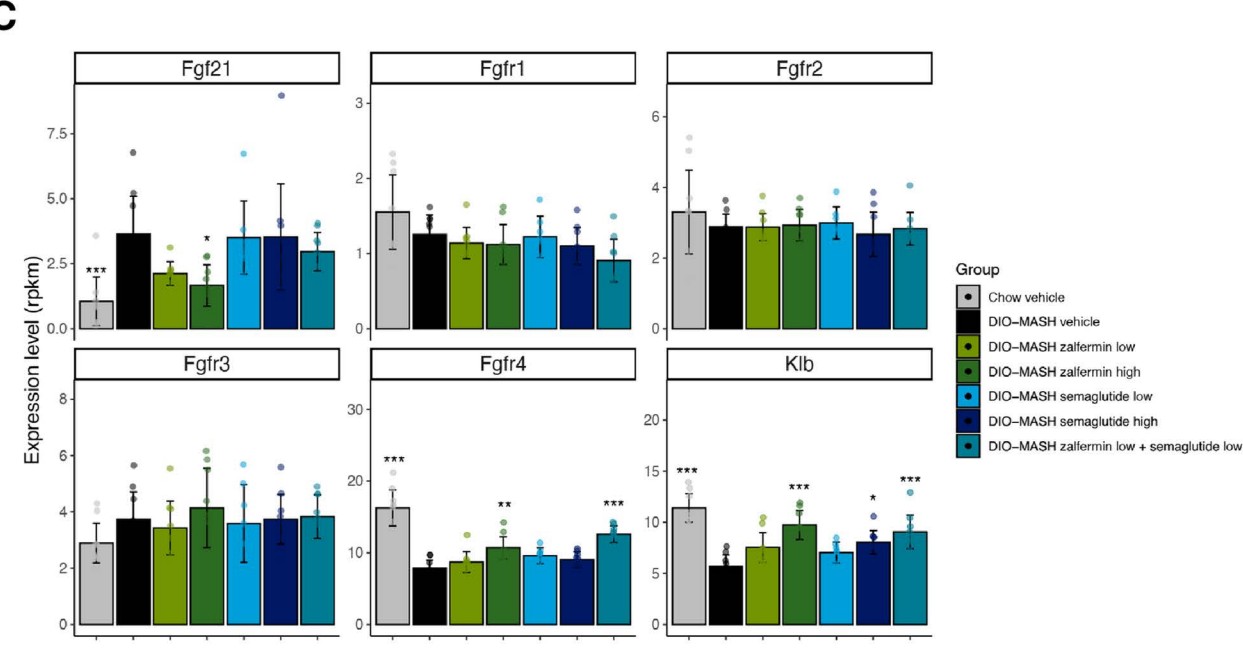

**Fig 5. Overall changes in liver transcriptome signatures following zalfermin and semaglutide treatment. (A)** PCA of samples based on 500 most variable expressed genes in the dataset. **(B)** Total number of differentially expressed genes compared with vehicle-dosed DIO-MASH mice. **(C)** Expression of *Fgf21* and canonical *Fgf* receptors. AMLN DIO-MASH mice showed significantly increased *Fgf21* expression and reduced *Ffgr4* and *Klb*

expression compared with chow-fed controls. Low-dose zalfermin-semaglutide combination treatment as well as high-dose zalfermin monotherapy partially reversed *Ffgr4* and *Klb* expression. Abbreviations: AMLN, amylin liver nonalcoholic steatohepatitis; DEG, differentially expressed gene; DIO, diet-induced obesity; MASH, metabolic dysfunction–associated steatohepatitis; PCA, principal component analysis. *$p < 0.05$, **$p < 0.01$, ***$p < 0.001$ vs. DIO-MASH vehicle.

fibrosis stage F2–F3 [30]. There is also an ongoing phase 2 trial that is evaluating the effect of zalfermin and semaglutide combination therapy in patients with MASH with fibrosis stage F2–F3 or F4c (NCT05016882). Other investigational long-acting FGF21 analogs have recently been profiled in patients with MASH. A 24-week phase 2b clinical trial evaluating the Fc-conjugated FGF21 analog efruxifermin (AKR-001) indicated significant improvements in NAS and fibrosis stage in patients with MASH with fibrosis stage F2–F3 compared with placebo [8]. Interestingly, efruxifermin has also recently been shown to provide clinically meaningful reductions in liver fat (hepatic fat fraction) and non-invasive markers of fibrosis in patients with MASH and type 2 diabetes already on GLP-1RA treatment [18]. In contrast, a PEGylated FGF21 analog pegbelfermin (BMS-986036), administered once weekly did not meet FDA-accepted primary histological endpoints (≥1-point decrease in fibrosis score without worsening of MASH; MASH resolution without fibrosis worsening) in 24- to 48-week phase 2b studies in patients with MASH with fibrosis stage F3 or F4c [10,31]. In a 24-week phase 2b study, the glycoPEGylated FGF21 analog pegozafermin (BIO89–100) given once weekly improved fibrosis without worsening of MASH in 26% vs. 7% in the placebo group and resolved MASH without worsening of fibrosis in 37% vs. 2% in the placebo group [9].

Marginal weight loss or no clinically relevant change in body weight has been reported for this drug class in patients with MASH [8,9,31]. This is in line with the almost weight-neutral effects of long-acting FGF21 analogs, including efruxifermin and BOS-580, in individuals with obesity with or without type 2 diabetes [11,32]. The clinical reports are in contrast to the consistent weight loss (up to 20%) in mice, pigs, and nonhuman primates treated with FGF21 analogs [7], indicating nontranslatable body weight regulatory effects of FGF21 analogs in preclinical species. Despite similar efficacy on body weight loss, cumulative food intake was significantly decreased by high-dose semaglutide and increased by high-dose zalfermin monotherapy, indicating highly different modes of action on body weight. Therefore, the beneficial effects of high-dose zalfermin on the liver appear to occur even with increased intake of the MASH-inducing AMLN diet. Although we did not assess food preference in the present study, central FGF21 action has been shown to shift macronutrient preference toward protein sources [33]. We therefore speculate that enhanced food intake could reflect an increased drive for protein consumption rather than stimulating appetite function. Increased protein intake has been associated with weight loss and improved body composition which overall contributes to improved metabolic health [34,35]. This may suggest that FGF21-induced changes in nutrient preference could support beneficial metabolic adaptations despite increased caloric intake. In mice and non-human primates on a high-fat diet, FGF21 analogs have been shown to drive energy expenditure, so despite increases in food intake, body weight loss is observed [6,36]. However, it is still unknown if increases in energy expenditure are observed in humans treated with FGF21 analogs. Other FGF21 analogs have shown similar effects on food intake in DIO and DIO-MASH mouse models [29]. The disparity is therefore most likely attributed to species differences in the regulation of energy expenditure. In rodents, FGF21-stimulated energy expenditure is mediated via sympathomimetic stimulatory effects on adipose tissue thermogenesis through direct FGF21 central action to achieve negative energy balance [37]. While the comprehensive understanding of the neural circuits recruited by FGF21 is needed, hypothalamic KLB signaling could play a critical role in mediating FGF21 effects on energy homeostasis [37].

To fully harness the therapeutic potential of FGF21 analogs in the management of MASH, it could be desirable to combine FGF21 analog therapy with an anti-obesity drug. Given the added benefits of combined low-dose zalfermin and semaglutide treatment, we profiled the liver transcriptome with the aim to pinpoint molecular mechanisms potentially

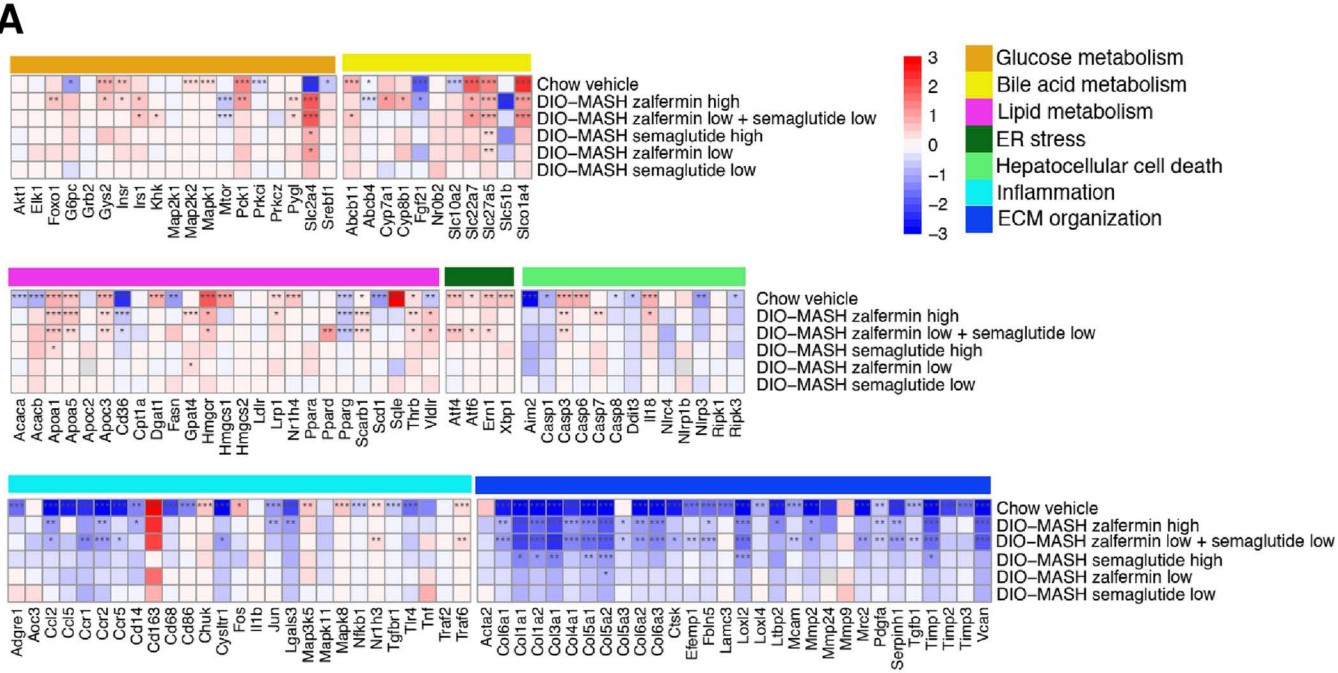

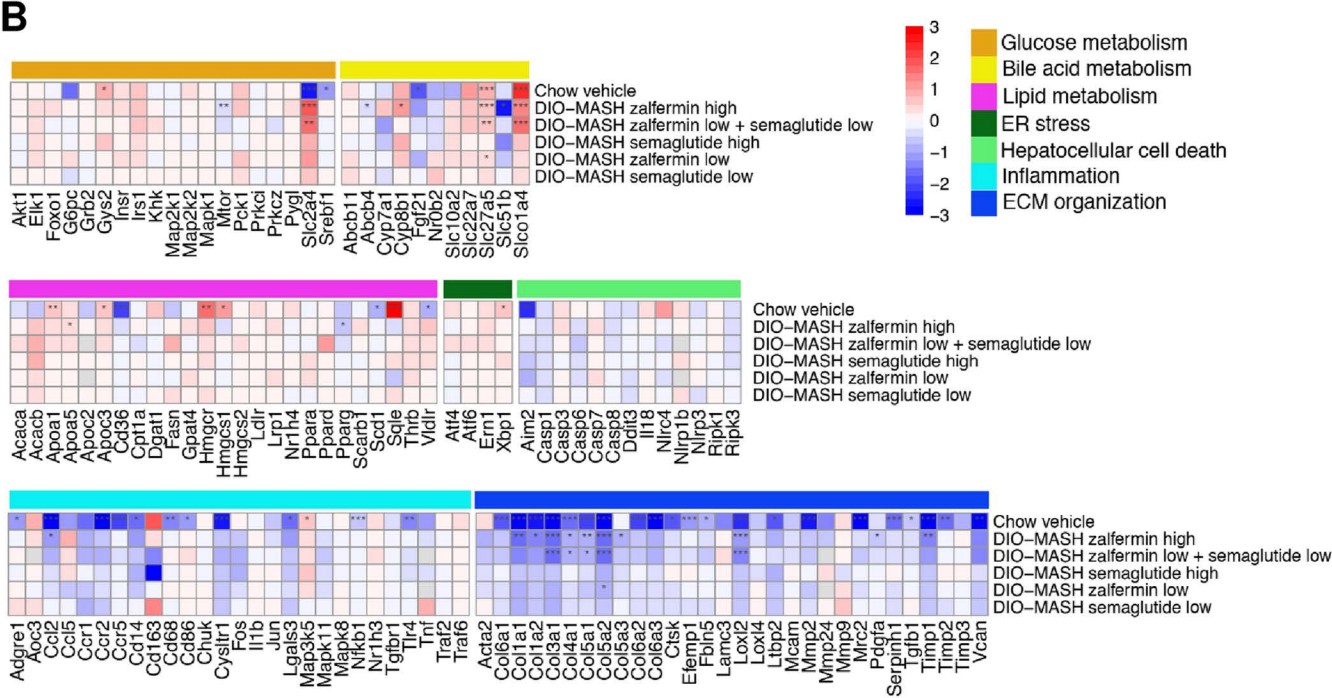

**Fig 6. Zalfermin and semaglutide effects on MASH and fibrosis candidate genes before and after correction for body weight loss.** Heatmaps illustrating alterations in the expression of candidate genes associated with MASH and fibrosis (log$_2$-fold change vs. DIO-MASH vehicle controls) without **(A)** or with **(B)** correction for changes in body weight. Combined low-dose zalfermin-semaglutide treatment demonstrated body weight–independent suppressive effects on fibrosis-associated gene expression programs. Red and blue signify upregulated and downregulated gene expression, respectively. Abbreviations: DIO, diet-induced obesity; ECM, extracellular matrix; ER, endoplasmic reticulum; MASH, metabolic dysfunction–associated steatohepatitis. *$p < 0.05$, **$p < 0.01$, ***$p < 0.001$ vs. DIO-MASH vehicle.

contributing to the observed strong effects on metabolic, biochemical, and histological markers in AMLN DIO-MASH mice. By comparing the effect of individual monotherapies on hepatic global gene expression signatures, it was evident that both compounds exhibited strong effects on gene expression markers of lipid handling, bile acid metabolism, and ECM organization, which were accentuated upon low-dose combination therapy. The low-dose combination treatment and high-dose zalfermin monotherapy were found to have a significant effect on gene markers of lipid metabolism and ECM organization even when adjusting for body weight, inviting the possibility that additive effects of combined low-dose zalfermin-semaglutide therapy were partially driven by molecular mechanisms unrelated to weight loss. Similar to patients with MASH and obese individuals who exhibit increased circulating FGF21 levels, an increase in plasma FGF21 concentrations has been reported in DIO mice, which also exhibit FGF21 resistance due to desensitization of FGFR1/KLB signaling [38]. It is therefore possible that the weight loss promoted by semaglutide could reverse FGF21 resistance, thereby enhancing efficacy of zalfermin. In support of this notion, low-dose combination treatment and high-dose zalfermin monotherapy partly reversed hepatic *Klb* downregulation in AMLN DIO-MASH mice. Also, semaglutide can improve FGF21 sensitivity in DIO mice by stimulating hepatic expression of FGFR1 and KLB [39]; furthermore, liraglutide has been shown to increase FGF21 sensitivity in adipose tissue by increasing expression of *Klb* and *Fgfr1−3* [40], indicating a great potential of GLP-1 agonism to increase FGF21 sensitivity.

Low-dose combination therapy partially reversed elevated plasma TC and liver TG levels in AMLN DIO-MASH mice. Consistent with a previous report, the AMLN diet did not induce hypertriglyceridemia, likely attributed to deficient hepatic TG secretion, as high dietary cholesterol levels lower endogenous cholesterol ester and lipoprotein synthesis [27]. Because zalfermin and semaglutide showed opposing effects on fat/cholesterol-rich AMLN diet intake, the benefits of zalfermin and semaglutide on these markers are likely due to different mechanisms. Zalfermin may improve plasma TC and liver TG levels by both direct and indirect (e.g., by inhibition of lipolysis and lowering of plasma insulin) liver action [41]. As hepatocytes do not express the GLP-1 receptors [42], semaglutide may potentially lower these lipid markers by a combination of reduced AMLN diet intake, decreased gastric emptying, and improved hepatic insulin sensitivity. Consistent with zalfermin and semaglutide targeting different molecular mechanisms, effects on liver metabolic transcriptome signatures were most pronounced for zalfermin. Key players in bile acid regulation, *Abcb11*, *Slc22a7*, *Slc27a5*, and *Slco1a4,* were upregulated by zalfermin. ABCB11 (also known as BSEP) is responsible for transporting bile acids from hepatocytes into the bile, while loss of Slc25a7 activates hepatic stellate cells and promotes fibrosis [43], potentially indicating a beneficial effect of zalfermin in MASH via bile acids regulation. Bile acids and their metabolites are toxic to hepatocytes, and bile acid synthesis is therefore tightly regulated by several negative feedback loops [44]. For example, bile acids activate hepatic farnesoid X receptor (FXR) signaling, which in turn stimulates the expression of the negative FXR co-regulators SHP and NR0B2, resulting in reduced expression of *Cyp7a1*, the major bile acid–producing enzyme. As *Cyp7a1* mRNA levels were increased by high-dose zalfermin, we speculate that zalfermin may improve hypercholesterolemia by stimulating cholesterol flux into bile acids. It should be noted that zalfermin-induced *Cyp7a1* expression was abolished by correcting for body weight. In addition, intestinal FXRs play an important role in controlling bile acid homeostasis, as ileal FXR activation increases the synthesis and release of FGF15/19 (FGF19 ortholog in mice), which directly acts on hepatocytes to repress bile acid synthesis and stimulate gallbladder filling [45]. It is therefore noteworthy that high-dose zalfermin and low-dose combination treatment partially reversed *Fgfr4* and *Klb* downregulation in AMLN DIO-MASH mice, potentially reflecting enhanced intrahepatic FGF15/19 activity. Although we did not assess bile acid levels in the present study, pegbelfermin has been reported to reduce circulating secondary bile acid levels in patients with MASH [46], supporting that FGF21 analogs can improve bile acid homeostasis in MASH. Zalfermin also consistently regulated markers of insulin sensitivity (upregulation of *Insr* and *Irs1,* downregulation of *Mtor*) and glycogen turnover (*Gys2, Pygl*), pointing toward overall improved hepatic glucose homeostasis.

## Conclusions

Zalfermin and semaglutide have added therapeutic effects in a translational mouse model of biopsy-confirmed MASH, supporting their combined use at low doses to improve treatment outcomes in MASH. Notably, high-dose zalfermin alone achieved comparable efficacy to the low-dose combination across most endpoints. Both low-dose combination and high-dose zalfermin showed an effect on hepatic ECM signatures, even after adjusting for body weight, supporting the clinical development of zalfermin as monotherapy and combination therapy.

## Supporting information

**S1 Fig. Zalfermin and semaglutide have opposing effects on real-time food intake in AMLN DIO-MASH mice. (A)** Daily food intake monitored during treatment week 1–2 and thereafter once weekly. **(B)** Daily cumulative food intake (g and kcal, treatment week 1–2). **(C)** Total food intake (g and kcal, treatment week 1–2). ***$p < 0.001$ vs. DIO-MASH vehicle; #$p < 0.05$, ###$p < 0.001$ vs. Zal low + Sema low.
(TIF)

**S2 Fig. Comparison of individual pre-post liver biopsy histopathological scores. (A)** NAS (NAFLD Activity Score), **(B)** fibrosis stage, **(C)** steatosis score, **(D)** lobular inflammation score, (E) hepatocyte ballooning degeneration score in AMLN DIO-MASH mice (n = 11–12 per group) administered vehicle (SC, QD), zalfermin 0.05 mg/kg (Zal low), zalfermin 0.2 mg/kg (Zal high), semaglutide 3 μg/kg (Sema low), semaglutide 120 μg/kg (Sema high), or zalfermin 0.05 mg/kg + sema-glutide 3 μg/kg (Zal low + Sema low) for 8 weeks. Vehicle-dosed (SC) chow-fed mice served as normal controls (n = 10).
(TIF)

**S3 Fig. Representative photomicrographs showing improvement in quantitative markers of inflammation (CD11b, Gal-3) and fibrogenesis (α-SMA), whereas fibrosis (Col1a1) appears unchanged after low-dose combination therapy with zalfermin and semaglutide.** AMLN DIO-MASH mice were administered vehicle (SC, QD), zalfermin 0.05 mg/kg (Zal low), zalfermin 0.2 mg/kg (Zal high), semaglutide 3 μg/kg (Sema low), semaglutide 120 μg/kg (Sema high), or zalfermin 0.05 mg/kg + semaglutide 3 μg/kg (Zal low + Sema low) for 8 weeks. Vehicle-dosed (SC) chow-fed mice (Chow Vehicle) served as normal controls. Magnification 20 ×; scale bar, 100 μm.
(TIF)

**S4 Fig. Differentially expressed genes (DEGs) in DIO-MASH mice. (A)** Upset plot depicting DEGs that are unique and shared among the DIO-MASH treatment groups. **(B)** As A, after correcting for body weight. **(C)** Venn diagram depicting overlapping and distinct differential expression of genes in groups receiving low-dose combination treatment or monother-apy with zalfermin and semaglutide compared with vehicle controls. **(D)** Venn diagram depicting overlapping and distinct differential expression of genes in groups receiving low-dose combination treatment or high-dose monotherapy with zalfer-min and semaglutide compared with vehicle controls.
(TIF)

**S5 Fig. Total number of differentially expressed genes (DEGs) compared with vehicle-dosed DIO-MASH mice with correction for changes in body weight.**
(TIF)

**S1 Data. Raw data PLoS One.**
(XLSX)

## Acknowledgments

Editorial support was provided by Diana Marouco, PhD, and Liam Gillies, PhD, of Apollo, OPEN Health Communications (London, UK) in accordance with Good Publication Practice (GPP) guidelines (www.ismpp.org/gpp-2022).

## Author contributions

**Conceptualization:** Jenny Norlin, Elisabeth D. Galsgaard, Michael Feigh, Sanne S. Veidal, Markus Latta, Emma Henriksson, Birgitte Andersen.

**Formal analysis:** Jenny Norlin, Maria Dermit, Nikos Sidiropoulos, Elisabeth D. Galsgaard, Sanne S. Veidal, Markus Latta, Emma Henriksson, Birgitte Andersen.

**Writing – review & editing:** Jenny Norlin, Maria Dermit, Henrik H. Hansen, Emma Henriksson, Birgitte Andersen.

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
