## [Decision Letter · Decision Letter 0]

7 Apr 2025

Dear Dr. Andersen,

Thank you for submitting your manuscript to PLOS ONE. After careful consideration, we feel that it has merit but does not fully meet PLOS ONE’s publication criteria as it currently stands. Therefore, we invite you to submit a revised version of the manuscript that addresses the points raised during the review process.

We look forward to receiving your revised manuscript.

Kind regards,

Habib Yaribeygi

Academic Editor

PLOS ONE

Journal Requirements:

3. Thank you for stating the following financial disclosure: [Funded by Novo Nordisk A/S, in accordance with Good Publication Practice (GPP) guidelines (www.ismpp.org/gpp-2022).]. 

4. Thank you for stating the following in the Competing Interests: [J.N., M.D., N.S., E.D.G., S.S.V., M.L, E.H. and B.A. are employees and shareholders of Novo Nordisk A/S. H.H.H. and M.F. are employees and shareholders of Gubra]. 

We note that you received funding from a commercial source: [Novo Nordisk]

Within this Competing Interests Statement, please confirm that this does not alter your adherence to all PLOS ONE policies on sharing data and materials by including the following statement: ""This does not alter our adherence to PLOS ONE policies on sharing data and materials.” (as detailed online in our guide for authors http://journals.plos.org/plosone/s/competing-interests).  If there are restrictions on sharing of data and/or materials, please state these. Please note that we cannot proceed with consideration of your article until this information has been declared.

5. We note that your Data Availability Statement is currently as follows: [All relevant data are within the manuscript and its Supporting Information files.]

6. Please amend either the title on the online submission form (via Edit Submission) or the title in the manuscript so that they are identical.

Reviewers' comments:

Reviewer's Responses to Questions

**Comments to the Author**

1. Is the manuscript technically sound, and do the data support the conclusions?

Reviewer #1: Yes

Reviewer #2: Yes

2. Has the statistical analysis been performed appropriately and rigorously?

Reviewer #1: Yes

Reviewer #2: Yes

3. Have the authors made all data underlying the findings in their manuscript fully available?

Reviewer #1: Yes

Reviewer #2: Yes

4. Is the manuscript presented in an intelligible fashion and written in standard English?

Reviewer #1: Yes

Reviewer #2: Yes

Reviewer #1: This preclinical study investigated the effects of zalfermin, a novel long-acting FGF21 analogue, both alone and in combination with semaglutide, in the AMLN DIO-MASH mouse model—a well-established, biopsy-confirmed model of MASH and fibrosis. The findings highlight the additive benefits of a combined low-dose zalfermin and semaglutide treatment, which led to metabolic, biochemical, and histological improvements comparable to or exceeding those observed with high-dose monotherapies. Hepatic transcriptome analyses suggest that zalfermin’s effects, both as a monotherapy and in combination, were partially mediated by weight loss–independent molecular mechanisms.

The data indicate that treatment improved liver metabolic markers, inflammation, and ER stress pathways, with these changes largely attributable to weight loss. However, a subset of ECM-related genes remained suppressed after adjusting for body weight in DIO-MASH mice, while PSR staining results suggested no significant improvement in liver fibrosis. Given the growing interest in FGF21 and incretin-based therapies for MASH, this study’s focus on the combination of an FGF21 analogue with semaglutide is particularly relevant to a broad audience. The manuscript is well-written and the data well-organized, with only a few minor comments:

1. In Fig. 1C, the Y-axis unit should be "%" instead of "g."

2. Some scale bars are difficult to see; please correct them, such as in the right panels of Fig. 3 and Fig. S3.

3. In Fig. S1B, label the group information.

Reviewer #2: Really interesting article and a pleasure to read. The study is well-executed, and you successfully integrate different aspects—pharmacotherapy, genetics, and histology. It makes me curious about how humans would respond to this protocol.

I have just four remarks:

1) The abstract would be more easy to read if you add the sections "background, methods, results, conclusion"

2) There is a typo in the title: "has additive therapeutic" instead of "hasadditivetherapeutic"

3) In the conclusion, you should emphasize more strongly that combination therapy is effective for both weight loss and histological changes. Otherwise, one might be inclined to think that a high dose of either zalfermin or semaglutide could be an alternative.

4)How do you explain the increased food intake observed with zalfermin while still achieving weight loss? In your discussion, you mention the preference for protein sources, but mice still consume more overall.

**Do you want your identity to be public for this peer review?** For information about this choice, including consent withdrawal, please see our Privacy Policy

Reviewer #1: No

Reviewer #2: **Yes: ** Willy Theel

---

## [Author Response · Author response to Decision Letter 1]

20 May 2025

REBUTTAL LETTER

Dear editor,

Thank you for the swift and positive handling of our manuscript entitled “The combination of zalfermin and semaglutide has additive therapeutic effects in a diet-induced obese and biopsy-confirmed mouse model of MASH.” We sincerely appreciate the reviewers’ insightful and constructive comments, which have helped us improve the manuscript.

We have carefully addressed each point raised and provided our detailed responses below (indicated in blue italics). Revisions in the manuscript are highlighted in yellow for clarity.

We believe the editorial process has significantly strengthened the manuscript, and we hope it is now suitable for publication.

As requested, please see the amended statements below.

Competing Interests Statement: The study was funded by Novo Nordisk A/S. The funder was involved in the study design, data collection and analysis, decision to publish, and preparation of the manuscript. This does not alter our adherence to PLOS ONE policies on sharing data and materials.

Data Availability Statement: All relevant data are within the manuscript and its Supporting Information files. The RNA sequencing data are available in the Gene Expression Omnibus (GEO, https://www.ncbi.nlm.nih.gov/geo/; accession number GSE256063). The code used to generate the results is available in the GitHub repository (https://github.com/novonordisk-research/Zalfermin_liver_RNAseq).

On behalf of all co-authors,

Yours sincerely,

Birgitte Andersen, Scientific Vice President,

Diabetes, Obesity & MASH, Novo Nordisk

REVIEWER 1

This preclinical study investigated the effects of zalfermin, a novel long-acting FGF21 analogue, both alone and in combination with semaglutide, in the AMLN DIO-MASH mouse model, a well-established, biopsy-confirmed model of MASH and fibrosis. The findings highlight the additive benefits of a combined low-dose zalfermin and semaglutide treatment, which led to metabolic, biochemical, and histological improvements comparable to or exceeding those observed with high-dose monotherapies. Hepatic transcriptome analyses suggest that zalfermin’s effects, both as a monotherapy and in combination, were partially mediated by weight loss independent molecular mechanisms.

The data indicate that treatment improved liver metabolic markers, inflammation, and ER stress pathways, with these changes largely attributable to weight loss. However, a subset of ECM-related genes remained suppressed after adjusting for body weight in DIO-MASH mice, while PSR staining results suggested no significant improvement in liver fibrosis. Given the growing interest in FGF21 and incretin-based therapies for MASH, this study’s focus on the combination of an FGF21 analogue with semaglutide is particularly relevant to a broad audience. The manuscript is well-written and the data well-organized, with only a few minor comments:

1. In Fig. 1C, the Y-axis unit should be "%" instead of "g."

2. Some scale bars are difficult to see; please correct them, such as in the right panels of Fig. 3 and Fig. S3.

3. In Fig. S1B, label the group information.

Response: Thank you for bringing these issues to our attention. The typographical error has been corrected (Fig. 1C), scale bars have been enhanced for better visibility (Fig. 3, Fig. S3), and legends have been added where needed (Fig. S1B).

REVIEWER 2

Really interesting article and a pleasure to read. The study is well-executed, and you successfully integrate different aspects, pharmacotherapy, genetics, and histology. It makes me curious about how humans would respond to this protocol. I have just four remarks:

1) The abstract would be more easy to read if you add the sections "background, methods, results, conclusion"

Response: Although the abstract is structured for clarity, we note that PLOS ONE does not permit subheadings in abstracts.

2) There is a typo in the title: "has additive therapeutic" instead of "has additive therapeutic"

Response: The manuscript title reads ‘The combination of zalfermin and semaglutide has additive therapeutic effects in a diet-induced obese and biopsy-confirmed mouse model of MASH’. We have reviewed it carefully and did not identify any typographical errors.

3) In the conclusion, you should emphasize more strongly that combination therapy is effective for both weight loss and histological changes. Otherwise, one might be inclined to think that a high dose of either zalfermin or semaglutide could be an alternative.

Response: We thank the reviewer for this insightful comment. In our preclinical study, the low-dose combination of zalfermin and semaglutide significantly improved both metabolic parameters and liver histology in MASH. However, since high-dose zalfermin alone demonstrated comparable efficacy across most endpoints, these findings also support the potential of zalfermin as a monotherapy. We have revised the Conclusion section to reflect both therapeutic options (p. 28).

4) How do you explain the increased food intake observed with zalfermin while still achieving weight loss? In your discussion, you mention the preference for protein sources, but mice still consume more overall.

Response: We speculate that zalfermin-induced changes in nutrient preference could support beneficial metabolic adaptations despite increased caloric intake. Increased protein intake has been associated with fat mass loss which contributes to reduced body weight and improved metabolic health. This aspect is now considered in the Discussion (p.25).

---

## [Decision Letter · Decision Letter 1]

20 Aug 2025

The combination of zalfermin and semaglutide has additive therapeutic effects in a diet-induced obese and biopsy-confirmed mouse model of MASH

PONE-D-25-05768R1

Dear Dr. Andersen,

We’re pleased to inform you that your manuscript has been judged scientifically suitable for publication and will be formally accepted for publication once it meets all outstanding technical requirements.

Kind regards,

Nobuyuki Takahashi, Ph.D.

Academic Editor

PLOS ONE

Additional Editor Comments:

I apologize for the delay in the review process. Although one reviewer provided no response regarding a minor revision, I have decided to accept your revision as an academic editor.

Reviewers' comments:

Reviewer's Responses to Questions

**Comments to the Author**

Reviewer #1: All comments have been addressed

2. Is the manuscript technically sound, and do the data support the conclusions?

Reviewer #1: Yes

3. Has the statistical analysis been performed appropriately and rigorously?

Reviewer #1: Yes

4. Have the authors made all data underlying the findings in their manuscript fully available?

Reviewer #1: Yes

5. Is the manuscript presented in an intelligible fashion and written in standard English?

Reviewer #1: Yes

Reviewer #1: (No Response)

**Do you want your identity to be public for this peer review?** For information about this choice, including consent withdrawal, please see our Privacy Policy

Reviewer #1: No

---

## [Editor Report · Acceptance letter]

PONE-D-25-05768R1

PLOS ONE

Dear Dr. Andersen,

I'm pleased to inform you that your manuscript has been deemed suitable for publication in PLOS ONE. Congratulations! Your manuscript is now being handed over to our production team.

Kind regards,

on behalf of

Dr. Nobuyuki Takahashi

Academic Editor

PLOS ONE